# The stigmatization of mental illness by mental health professionals: Scoping review and bibliometric analysis

**Michael Jauch**[1]*, **Stefano Occhipinti**[1,2], **Analise O'Donovan**[1]

**1** School of Applied Psychology, Griffith University, Brisbane, Queensland, Australia, **2** Department of English and Communication, International Research Centre for the Advancement of Health Communication, Hong Kong Polytechnic University, Hong Kong, China

* michael.jauch@griffithuni.edu.au

**Data Availability Statement:** All relevant data are within the paper and its Supporting information files.

**Funding:** The author(s) received no specific funding for this work.

## Abstract

Although research suggests that mental health professionals stigmatize mental illness, studies on this topic are relativity new. Little is known about the state of this research and existing literature reviews exhibit multiple limitations. Accordingly, a scoping review was performed on the endorsed stigmatization of mental illness by mental health professionals, with the aim of exploring how research is conducted and whether there are gaps in the literature. Studies were included from any time period if they supplied findings on the endorsed stigmatization of mental illness by mental health professionals. Research was identified through electronic databases (i.e., PsycINFO, Embase, Medline, Scopus) and other sources (i.e., the Griffith University library, Google Scholar, literature reviews). It was found that the research is characterized by a number of limitations, and little progress has been made in this important domain. Among other limitations, there was a lack of comprehensive studies on the relative stigma of mental illness and how the components of stigmatization relate to each other. A bibliometric analysis also found that a large proportion of the research is not connected by references. Recommendations were made with respect to future research in this area.

## Introduction

Studies suggest that health professionals can be just as susceptible to stigmatizing attitudes within their respective fields as the general population [1–3]. Researchers have found both that physiotherapists stigmatize individuals who are overweight and physicians stigmatize lung cancer patients [2–4]. Of primary relevance to the current paper, provider-based stigma is of significant concern in the domain of mental health, where mental health professionals have been found to express stigmatizing reactions towards people with mental illness [1, 5–7]. For example, a study by Servais and Saunders [8] found that clinical psychologists perceive people with borderline features to be dangerous, and people with schizophrenia to be undesirable. This is troubling considering that the stigmatization of mental illness can have a negative impact on people who suffer from mental illness, and provider-based stigma is associated with negative outcomes for health care recipients [3, 9–11]. The stigmatizing of mental illness

**Competing interests:** The authors have declared that no competing interests exist.

including that by mental health professionals is also increasingly recognized as a public health concern and there are calls for government-based responses in some jurisdictions [12]. Thus, there is strong impetus for research that can provide insight into how the stigmatization of mental illness by mental health professionals can be mitigated. Surprisingly, studies on mental health professionals stigmatizing mental illness are fairly new and only a relatively limited amount of research has been conducted on this subject [1, 5–7]. As well, the state of research on mental health professionals stigmatizing mental illness is mostly unknown and extant literature reviews on the topic add little clarity to this and are marked by several limitations [1, 5–7, 13]. Therefore, it is timely for the state of research on mental health professionals stigmatizing mental illness to be investigated in a rigorous manner. Accordingly, a scoping review will be undertaken, as this type of literature review serves to describe the state of research in an area, including how studies are conducted and whether there are any gaps in the literature [14–16].

## The context

A large number of studies have demonstrated that mental illness is stigmatized by the general population [1, 10, 11, 17]. *Stigmatization* is a collective system of negative reactions that are elicited by human attributes [18–20]. The components of stigmatization include negative *stereotypes* (e.g., incompetence), negative *emotions* (e.g., anger), and *discrimination* (e.g., avoidance) [18, 19]. One variant of stigmatization is *endorsed stigma*, referring to expressed agreement with stigmatizing reactions [20]. When endorsed stigma manifests as discrimination against individuals with mental illness, these people encounter a range of negative consequences [1, 5, 9]. For example, individuals with mental illness experience limited access to housing, unemployment, financial difficulties, health problems, and poor treatment outcomes [5, 10, 11, 17]. In fact, such consequences are in themselves evidence of *structural stigma*. Thus, mental illness causes harm directly, but such harm is also potentiated by the negative consequences associated with stigmatization [21].

Research suggests that in addition to the general population, mental health professionals also stigmatize mental illness, and this has implications for public health and policy [1, 3, 5, 6, 12]. *Provider-based stigma* occurs when occupational groups endorse the stigmatization of the people they are meant to help and is related to negative repercussions for health care recipients [3, 20]. In the area of mental health, studies indicate that mental health professionals such as clinical psychologists endorse many of the same negative stereotypes, emotions, and behaviors as the general population [22–27]. This is of concern, as the stigmatization of mental illness is in many ways incompatible with good mental health practice. Given that provider-based stigma can have a negative impact on health care recipients, it is likely that the stigmatization of mental illness by mental health professionals is also linked to negative consequences for those who suffer from mental illness. Recently in Australia, the Productivity Commission exhibited an awareness of these likely negative outcomes in recommending to the government that action be taken towards reducing the stigmatization of mental illness by health professionals, which includes mental health professionals [12]. This is an example of institutions becoming more conscious of mental health professionals stigmatizing mental illness, and advocating for a change.

## Rationale

Despite the noted importance of research on mental health professionals stigmatizing mental illness, this topic has only more recently garnered attention from the scientific community, and seemingly only a small body of literature has accumulated in this area [1, 5–7, 13].

Research on mental health professionals stigmatizing mental illness appears to have emerged in the 2000s, with few studies being conducted prior to this time period [1, 5–7, 13]. To our knowledge, by 2014 only about 20 studies had been completed on mental health professionals stigmatizing mental illness [7]. In comparison to research with mental health professionals, the stigmatization of mental illness by the general population had been investigated in a number of studies prior to the 2000s, with some research being conducted as far back as the 1950s [21, 28–30]. Further, the state of research on mental health professionals stigmatizing mental illness is currently unknown, as extant literature reviews in this domain mostly summarize findings, and provide little to no information about how research is being conducted and whether there are any gaps in the literature [1, 5–7, 13].

Existing literature reviews on mental health professionals stigmatizing mental illness are also limited in ways that have likely caused literature to be overlooked and make it difficult to draw conclusions about the research area [1, 5–7, 13]. First, not all of these reviews were conducted with systematic and reproducible methods for identifying relevant studies. Further, the literature reviews that did satisfy this level of rigor, arguably did not use search terms with sufficient detail to capture most of the literature on mental health professionals stigmatizing mental illness. For the most part, the search terms that were used included two or three variants of the terms *stigmatization*, *mental illness*, and *mental health professional*, respectively. Although these terms outline the relevant articles generally, they may exclude studies that are more specific, such as those that could be on particular mental disorders (e.g., schizophrenia, major depression, alcohol use disorder), or particular mental health professions (e.g., psychologists, psychiatric nurses, counselors). There are also several inconsistencies between the literature reviews on mental health professionals stigmatizing mental illness. One inconsistency regards the focus of the literature reviews. Specifically, none of the reviews summarized research on just the endorsed stigmatization of mental illness by mental health professionals (e.g., one review included studies on stigmatization by the general population), and the reviews differed in the extent to which this was the focus. Possibly related to this, there are inconsistencies between some of the literature reviews with respect to the number of articles included on the endorsed stigmatization of mental illness by mental health professionals. For instance, despite Wahl and Aroesty-Cohen [1] including 17 studies in their review, one year later a review by Ahmedani [13] only included three studies. The last inconsistency between the literature reviews concerns restrictions placed on time periods. Namely, Wahl and Aroesty-Cohen [1] did not review studies prior to 2004, Carrara et al. [6] only reviewed studies between 1992 and 2015, and the other reviews did not restrict searches to any time period.

As there are several inconsistencies between the literature reviews on mental health professionals stigmatizing mental illness, the literature on this topic itself may be inconsistent. Such inconsistencies are highly likely observable in the manner research is executed, and likely to result in multiple gaps in the literature. With respect to the latter, type of mental disorder is one factor that is of particular interest to the current review. This variable is referred to here as the *relative stigma* of mental illness, or the degree to which mental disorders are stigmatized compared to other mental disorders. Within research on the general population, a number of studies have explored and demonstrated the relative stigma of mental illness with a range of mental disorders [21, 31–35]. A second variable, or more accurately system of variables, that is salient to this scoping review, is the components of stigmatization as they relate to each other. This has been examined in research with the general population, including studies on how the components relate to each other in a complete framework, with emotions mediating the relationship between stereotypes and discrimination [32, 36–39]. Understanding the relative stigma of mental illness and how the components of stigmatization relate to each other is crucial to reducing the stigmatization of mental illness generally and in particular among mental

health professionals. Knowledge of which mental disorders are stigmatized more than others provides guidance for interventions on which approaches are appropriate for the different mental disorder stigmas, and which stigmas should be prioritized. Moreover, having a grasp of how the components of stigmatization relate to each other is necessary for uncovering the mechanisms of change interventions should target to be effective. Inconsistencies in the literature may also impede the identification of broad findings within the research, and could be due to a lack of referencing connections between articles and research being published in lesser-known journals.

## Objectives

This scoping review aims to examine the state of available research on the endorsed stigmatization of mental illness by mental health professionals. In addition to investigating the state of research broadly, this scoping review aims to more precisely elucidate how studies are being conducted and whether there are gaps in the literature, two common scoping review objectives [14, 15]. As well, the current study aims to appraise whether there are any clear findings in the literature on mental health professionals stigmatizing mental illness and explore some of the bibliometric features of the research in this area. *Bibliometrics* can be defined as the study of referencing patterns amongst the various forms of literature [40, 41], and bibliometric analyses are finding increasing use in the field of mental illness stigma [e.g., 42]. These analyses will supply an indication of referencing connections between studies and a summary of the types of journals that articles are being published in.

## Method

This scoping review was guided by the methodological framework developed by Arksey and O'Malley [15] and is reported as per the Preferred Reporting Items for Systematic Reviews and Meta-Analyses extension for scoping reviews (PRISMA-ScR) guideline (S16 Appendix) [43].

### Eligibility criteria

Articles were selected if they provided findings on the endorsed stigmatization of mental illness by mental health professionals. Literature was also included on mental health professionals perceiving the stigmatization of mental illness in other mental health professionals. As this perceived form of stigma likely reflects endorsed stigma, this literature was selected as indirect evidence of the endorsed stigmatization of mental illness by mental health professionals. Articles were only included if the participants were referred to as mental health professionals broadly, or if it was clear that particular professional groups were part of the sample. Providers were considered mental health professionals if they were professionals from the fields of psychology, counseling/psychotherapy, social work, occupational therapy, psychiatric nursing, psychiatry, or general/family/primary care medicine. Participants were also regarded as mental health professionals if they were unspecified physicians working in a primary care setting, or any type of nurse or physician working in a psychiatric facility.

Research was excluded on the stigmatization of individual psychological symptoms (e.g., self-harm, suicide, hallucinations) and subclinical behaviors (e.g., alcohol consumption), as these attributes alone do not constitute mental illness. Similar to this, literature was excluded on diagnostic processes (e.g., a symptom of major depressive disorder is anhedonia), and beliefs regarding society-level decisions about people with mental illness (e.g., involuntary hospitalization) that were not explicitly linked to stigmatization. These phenomena were excluded as they are not necessarily indicative of stigmatization, and rather may reflect an objective assessment. Research was also not included if it was on the psychometric properties of

stigmatization scales, if the results combined mostly items irrelevant to stigmatization with comparatively few stigmatization items, and if the analyses compared one sample to a completely different sample. Literature with these characteristics was excluded because such research either does not add to an understanding of mental health professionals stigmatizing mental illness or is not conducive to unambiguous conclusions. See Table 1 for the full inclusion and exclusion criteria.

## Search and information sources

To access relevant studies via electronic databases, a three-part search string was constructed. This search string consisted of variations of the terms stigmatization, mental illness, and mental health professional. In addition to these general terms, specific mental disorders and particular professions were incorporated into the search string, to ensure that articles on specific mental disorders and professions were included in this review. Given the infeasibility of accounting for every classified mental disorder in the search string, only certain mental disorders were listed. These classifications were mood disorder, substance use disorder, anxiety disorder, impulse control disorder, depression, and schizophrenia. The first four of these were included because research suggests that globally these mental disorders are the most prevalent [44]. Schizophrenia and depression were added based on the observation that these mental disorders are the most commonly specified mental disorders in the literature on mental health professionals stigmatizing mental illness [1, 5–7, 13]. Refer to Table 2 for the complete search string in the form of one database search. It is noted that the search string accounts for the inclusion of midwives, despite the decision to only include non-psychiatric nurses if they work

**Table 1. Inclusion and exclusion criteria.**

| Inclusion Criteria |
|---|
| • The endorsed stigmatization of mental illness by mental health professionals |
| • Mental health professionals perceiving the stigmatization of mental illness in other mental health professionals |
| • Published in English |
| • Published in a peer-reviewed journal |
| • From any time period |
| • Qualitative or quantitative |
| Exclusion Criteria |
| • The stigmatization of individual psychological symptoms |
| • The stigmatization of subclinical behaviors |
| • Diagnostic processes |
| • Beliefs regarding society-level decisions about people with mental illness |
| • The psychometric properties of stigmatization scales |
| • Results that combine mostly items irrelevant to stigmatization with comparatively few stigmatization items |
| • Analyses that compare one sample to a completely different sample |
| • First-person accounts of mental health professionals stigmatizing mental illness |
| • Mental health professionals perceiving the stigmatization of mental illness in the general population |
| • Mental health professionals stigmatizing mental illness in other mental health professionals |
| • Mental health professionals stigmatizing mental illness in themselves |
| • Trainee mental health professionals |
| • Inaccessible articles |
| • Conference proceedings, books, and literature reviews |
| • Any non-empirical documents |

**Table 2. Search query protocol and flow for PsycINFO.**

| |
|---|
| *#1* (stigma* OR attitudes OR stereotyp* OR prejudice OR discrimination) |
| *#2* (mental illness OR mental disorder OR mental disease OR mental health OR psychiatric illness OR psychiatric disorder OR psychological disorder OR psychological illness OR psychiatric disease OR psychological disease OR psychopathology OR abnormal psychology OR depression OR mood disorder OR schizophrenia OR substance OR anxiety OR impulse control) |
| *#3* (provider OR mental health professional OR mental health practitioner OR health professional OR health practitioner OR medical professional OR medical practitioner OR clinician OR psychologist OR therapist OR psychotherapist OR counsellor OR counselor OR psychiatrist OR general practitioner OR GP OR nurse OR occupational therapist OR social worker OR physician OR midwi*) |
| *#4* #1 AND #2 AND #3 |
| *#5* Limit #4 to (peer reviewed journal and English language) |

in a psychiatric facility. This discrepancy is due to the inclusion criteria being refined after the review process had progressed too far for changes to be made to the search string.

The search string was entered into a psychology database, two biomedical databases, and a multidisciplinary database. PsycINFO was the psychology database, Embase and Medline were the biomedical databases, and Scopus was the multidisciplinary database. Searches were carried out on the basis of titles, abstracts and keywords, and all searches occurred for the first time on the 29th of May 2019. These searches were replicated on either the 9th (PsycINFO, Embase) or 10th (Medline, Scopus) of December 2019 to identify any articles that had been added to the databases since the first set of searches. The same set of searches were also conducted for a third time on either the 16th (Scopus, Embase) or 17th (Medline, PsycINFO) of September 2021. In addition to the database searches, relevant studies were found through Google Scholar and the Griffith University library. As the full search string could not be entered into Google Scholar or the Griffith University library, a number of different searches had to be performed for these search engines. These searches involved combining terms from each of the three parts of the database search string (e.g., mental health professionals and mental illness stigma), and were executed roughly around the same time as the first database searches. Potentially relevant articles were also screened for in literature reviews on mental health professionals stigmatizing mental illness. These reviews were by Wahl and Aroesty-Cohen [1], Schulze [5], Henderson et al. [7], Carrara et al. [6], and Ahmedani [13]. After studies were located through the first set of database searches, search engines, and literature reviews, the reference lists of relevant studies were inspected to find any articles that were not acquired via the primary sources.

## Selection of sources of evidence

Relevant literature was identified by screening titles, abstracts, and full texts in a sequential order. In some cases, if the abstracts were either non-existent or too vague, articles were probed further without reading the full text. This phase of the scoping review was conducted by one reviewer. However, to verify that inclusion and exclusion was congruent with the eligibility criteria, an evaluation of inter-reviewer consistency was performed with a second reviewer. This first entailed randomly selecting a quantity of included and excluded literature that corresponded to 10% of the ultimate corpus of relevant articles (not including the second and third database searches). Half of this literature was comprised of included studies from any of the information sources, and the other half covered excluded articles from just the electronic databases. With the eligibility criteria in mind, the second reviewer then chose from this sample the literature that they deemed relevant and irrelevant while being blind to the

inclusion and exclusion decisions of the other reviewer. It was found that for 90% of the sampled literature the two reviewers made the same inclusion and exclusion choices, and this level of agreement was considered acceptable. For the remaining 10%, the two reviewers had a discussion about their decisions, and the eligibility criteria was adjusted where necessary. Previous inclusion and exclusion decisions were changed to be consistent with the new eligibility criteria.

## Data charting process and data items

As with the selection of sources of evidence, data charting was also executed by one reviewer. Data items were year of publication, countries the studies were conducted in, research methods, analytical approaches, populations sampled, measures of stigmatization, mental disorders included, predictors of stigmatization, component relations, and findings. To clarify, research methods denote the design of the studies, how mental illness was presented and whether an intervention was used, and analytical approaches include either the statistical or qualitative procedures employed to interpret data. Populations sampled refers to the types of mental health professionals that participated in the research, and any irrelevant populations (e.g., the lay population) that were not separated from the mental health professionals in the results. As a final clarification, findings represent the amount of stigmatization exhibited by the mental health professionals, and what factors do or do not account for variation in stigmatization, including interventions. For the bibliometric analysis, data were gathered on how frequently articles were cited in other articles, and on the rank (e.g., Q1) of the journals that the studies were published in.

## Synthesis of results and bibliometric analysis

Evidence was summarized via narrative form, tables, and illustrations (i.e., a flow-chart and citation network), and the bibliometric analysis followed Bhandari [45] and Donthu et al. [46]. To examine citation links between articles a *direct citation analysis* was performed with the program VOSviewer [47]. In order for VOSviewer to access as many articles as possible, several data sources were considered (i.e., bibliographic database files, supported application programming interfaces). Through this process it was established that compared to other sources Scopus provided access to the highest number of studies. As for the analysis itself, literature was read in through Scopus, citation was selected as the type of analysis, and documents were specified as the unit of analysis. This produced a citation network and a list of clusters. Each of the nodes within the citation network were weighted by the number of links to other nodes. Conclusions were drawn based on an inspection of the clusters and nodes. Data on the rank of the journals that the articles were published in were obtained through the webpage www.scimagojr.com, and data were collected by documenting the highest and most recent rank the journals had been given.

## Results

### Selection of sources of evidence

The first set of database searches yielded 9310 documents, and 116 articles were identified through other sources (i.e., search engines and literature reviews). These records were then combined, and after removing duplicates the remaining 5938 documents were screened for relevance. Through this, 184 articles were deemed relevant. The reference lists of these studies were then examined, and a further 89 studies were found that met the inclusion criteria. This brought the total number of relevant articles to 273. For the second set of database searches,

six new studies were found that fit the inclusion criteria, and for the third set of searches 21 new studies were identified. With these records, the final number of relevant articles was 300 (the complete process of selecting articles is summarized in Fig 1). In addition to excluding documents in accordance with the exclusion criteria, many records were excluded as they were either not focused on stigmatization (e.g., the effectiveness of a particular type of treatment, the self-efficacy and knowledge of mental health professionals), they were not about attitudes towards mental illness (e.g., attitudes towards evidence-based practice, attitudes towards people with HIV), or they were on stigmatization by the general population.

## Characteristics of sources of evidence and results of individual sources of evidence

Characteristics of sources of evidence and the results of individual sources of evidence are presented as supporting information.

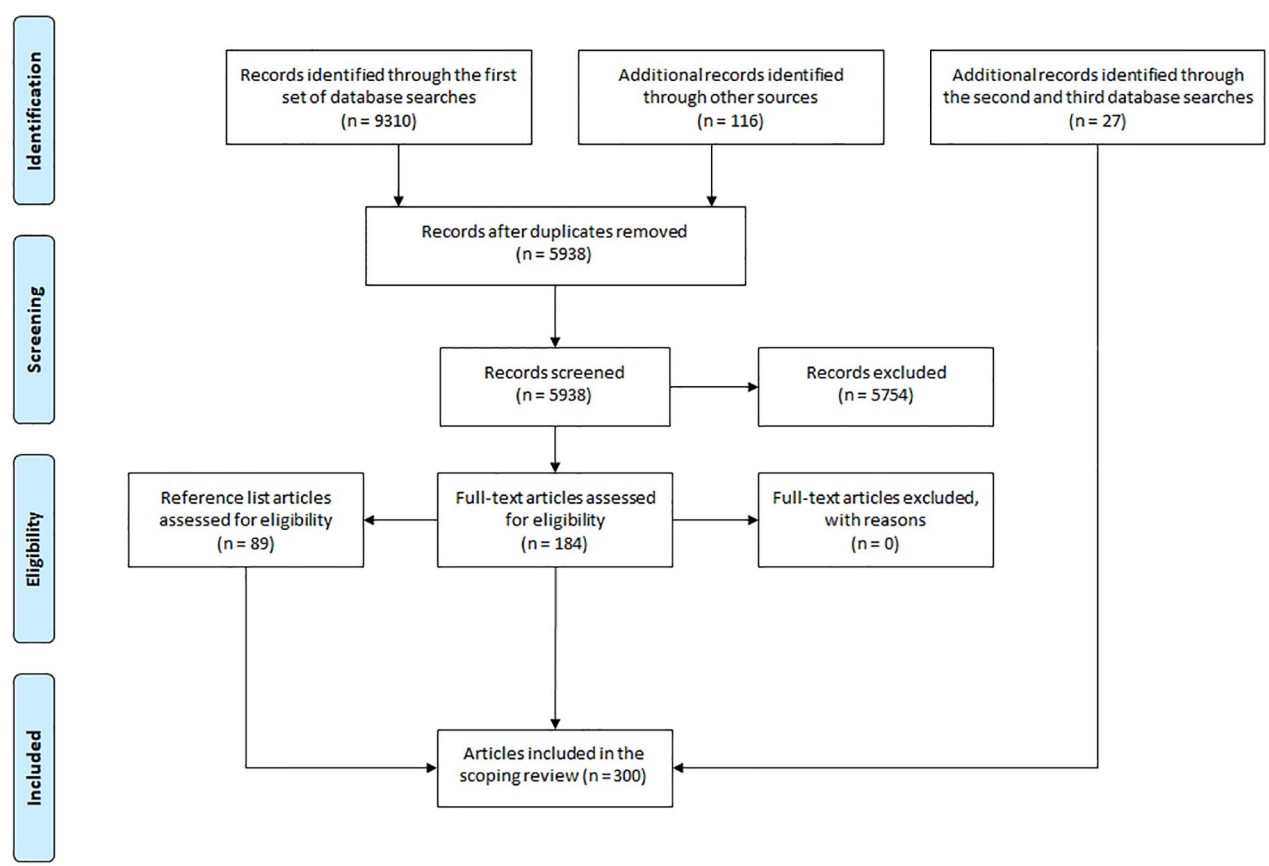

**Fig 1. PRISMA-ScR flow-chart.**

## Synthesis of results

**Preliminary findings.** On examining the 300 articles, a proportion were found to contain results that merged mental health professionals with irrelevant populations. This included members of the general population and in most cases non-mental health professionals (e.g., teachers, lawyers, engineers). In some instances, this occurred because non-mental health professionals (e.g., social work students, pharmacists, paramedics) were considered to be mental health professionals by the authors. Study findings that mix mental health professionals with irrelevant populations are problematic for the current review, as it is unclear whether these results truly reflect the mental health professionals sampled. Thus, for the remaining results, findings that merged mental health professionals with irrelevant populations, and article details that corresponded to this were excluded. Consequently, 34.33% of studies were completely excluded, 22% of articles were included but only a subset of the applicable characteristics could be used, and 43.33% of studies were included in full. This meant that 197 out of the original 300 articles were the basis of the following analyses.

Most of the studies were published in the 21$^{st}$ century, with 32.49% appearing in the 2000s, and 43.65% emerging in the 2010s and 2020s as a whole. Prior to this, 10.66% of articles were published in the 1990s, small quantities of studies appeared in the 1980s, 1970s, and 1960s, and just one article was published in the 1950s. Regarding geographical region, 30.96% of studies were conducted in North America, and most of this was in the United States. In comparison, only a few studies occurred in Brazil, and no other studies were conducted in South America. A further 25.38% of studies occurred in the United Kingdom, the majority of which were conducted in England. A very small number of studies were conducted in Scandinavia (e.g., Denmark, Norway), whereas 15.23% of studies occurred in other European regions (e.g., France, Italy, Germany). For the rest of the studies, 10.66% were conducted in the Middle East (e.g., Israel, Iran, Turkey), 9.14% occurred in Asia (e.g., China, Japan, India), and 7.61% were conducted in Africa (e.g., Nigeria, Ghana, South Africa). Also, 9.14% of studies occurred in Oceania, most of which were conducted in Australia.

**Research methods and analytical approaches.** The most common research method by far was the cross-sectional survey. This type of research method was utilized in 68.53% of studies, while only 4.06% and 6.09% of articles reported the use of longitudinal surveys and structured interviews, respectively. Experiments/quasi-experiments were employed in 11.68% of studies, and a few articles acknowledged the use of other quantitative methods (e.g., repertory grid technique, implicit association test, behavioral observation). Additionally, 6.60% of studies explored interventions for reducing stigmatization (e.g., alcoholism training program, Balint groups, educational workshop on borderline personality disorder). It may also be worth noting here that two of these interventions were not examined through the utilization of statistical modelling (e.g., test statistics, confidence intervals). Qualitative methods were considerably less common than quantitative methods. More specifically, 9.64% of articles reported the use of semi-structured interviews, and only a small number of unstructured interviews and focus groups were conducted. There was also one structured interview and four cross-sectional surveys that used open-ended questions to collect qualitative data. Across the research methods, mental illness was presented to participants via labels in 90.86% of studies, and for four more articles labels may have been employed but it was unclear. In contrast to labels, only 21.83% of studies made use of vignettes or some other means of portraying mental illness (e.g., short descriptions, audio recordings). Of these, vignettes were utilized the most, and in one article the mode of representing mental illness was not obvious.

A variety of quantitative analyses were used throughout the studies. Many of these were standard statistical procedures (e.g., correlation analysis, ANOVA, t-test, regression analysis)

and some were more rarely applied analyses (e.g., McNemar's test, Mantel-Haenszel test, Fisher's exact test). For several articles, the appropriate statistical figures were omitted to some extent, although more noteworthy was the observation that 89.61% of the applicable studies failed at least partially to correct for family-wise error rate. With respect to qualitative analyses, thematic analysis was employed in 51.35% of articles that included qualitative research. To a much lesser degree, a range of other qualitative analyses were utilized (e.g., contextual semantic interpretation, phenomenological analysis, discourse analysis), but not a single study used an analytical approach that involved making connections between themes (e.g., grounded theory). For both quantitative and qualitative analyses, some articles were not completely clear about the types of analyses, and 21.97% of studies that performed analyses were unable to be fully interpreted with the amount of information provided. These two issues occurred mostly for quantitative analyses.

**Types of mental health professionals.** Most of the prominent mental health professions were well represented within the corpus of studies. Psychiatrists (including registrars)/psychiatry professionals were in 31.98% of articles, general practitioners/family physicians/primary care physicians participated in 30.46% of studies, and social workers/social work professionals were included in 26.40% of articles. Adding to this, psychologists/psychology professionals were represented in 25.38% of articles, and psychiatric nurses participated in 20.30% of studies. However, counselors/psychotherapists were only included in 7.61% of articles, and occupational therapists/professionals from the field of occupational therapy participated in just 3.05% of studies. There was also one article that included general nurses that work in a psychiatric facility, and another study that included professionals from the field of neurology. To varying degrees, a number of studies were not clear about the types of mental health professionals that participated. Namely, 16.24% of articles included unspecified nurses from psychiatric facilities, 8.63% of studies included unspecified mental health professionals, and a small proportion of articles included unspecified physicians from either primary care or psychiatric settings.

**Measures and scales.** A large quantity of different measures were employed to examine stigmatization both across and often within studies. Of the measures, stereotypes were utilized the most, with 77.16% of articles reporting the use of this type of measure. A small amount of studies investigated stereotypes in general (e.g., an aggregate score of multiple stereotypes), but a plethora of specific stereotypes were also included throughout the articles. The five stereotypes that were used the most were about causal attributions (e.g., mental illness is caused by a lack of will power), prognosis (e.g., people with mental illness will not recover), dangerousness (e.g., people with mental illness are a danger to other patients and staff), difficulty (e.g., people with mental illness are difficult to treat), and incompetence (e.g., people with mental illness are incapable). None of the stereotypes took the form of perceived stigma.

Compared to stereotypes, emotions and behaviors were utilized far less to measure stigmatization. This was especially true for emotions, as only 23.35% of studies measured this component of stigmatization, whereas 40.10% of articles included behaviors as a measure of stigmatization. Similar to stereotypes, some studies explored emotions in general, yet a variety of particular emotions were included within the corpus of articles. The four most frequently measured emotions were fear, frustration, sympathy, and anger. For behaviors, general measures were also employed in several studies, although much fewer specific behaviors were examined in contrast to stereotypes and emotions. Avoidance (e.g., best to avoid people with this problem) and segregation (e.g., mental health facilities should be kept out of residential neighborhoods) were the two most commonly measured kinds of discrimination. Almost all of the emotions and behaviors reflected endorsed stigma, while just two articles reported perceived emotions (i.e., sympathy, empathy), and three studies measured perceived behavior (i.e., avoidance). On top of individual components of stigmatization, general stigmatization

(e.g., an aggregate score of different components of stigmatization, I dislike people with mental illness) was investigated in 44.16% of articles. Most of these studies measured endorsed and explicit general stigmatization, and perceived and implicit general stigmatization were included in only a few studies. Besides implicit general stigmatization, no other measures of implicit stigma were used in any of the articles (e.g., implicit stereotypes).

Stereotypes, emotions, behaviors, and stigmatization in general were either measured with established scales, or with scales that had not been previously validated. Amongst the studies a range of different established scales were utilized (e.g., Attribution Questionnaire, Medical Condition Regard Scale, Attitude to Personality Disorder Questionnaire) and the two most commonly used scales were the Community Attitudes towards Mental Illness questionnaire, and the Depression Attitude Questionnaire. In addition to the variety of such scales, the same scales were not always scored or computed in a consistent way across the articles (e.g., one study scoring the items on a 7-point Likert-type scale and another study scoring the items on a 4-point Likert-type scale; one article computing scores as a mean of all the items and another article using a total of all the items). Whether established scales were employed or not, there were some instances of measures being unclear (e.g., confirming behavioral responses was all the information provided), and in other cases, not enough information was supplied to interpret the measures in the results (e.g., when the meaning of points in a Likert-type scale were not specified). Further, several measures included items that are irrelevant to stigmatization (e.g., if depressed patients need antidepressants, they are better off with a psychiatrist than a GP), or items that do not match the construct supposedly being measured (e.g., taking care of borderline personality disorder patients can evoke unfamiliar feelings as an item for empathy).

**Types of mental illness.** Mental illness in general (e.g., the mentally ill, patients in a mental hospital, a person who has a mental illness) was included as stimuli for 43.65% of articles. Studies in the current review also elicited responses with a multitude of particular mental disorders (e.g., anxiety disorders, personality disorders, eating disorders). However, most of these mental disorders were present in only a small number of articles, and the majority of studies that presented specific categories of mental illness included the same three mental disorders. These were schizophrenia spectrum disorders, depressive disorders, and addiction/substance use disorders, and they were in 28.93%, 26.40%, and 17.77% of articles, respectively. Additionally, some studies included comorbid disorders, and several articles either did not specify the mental disorders or reported that the participants reacted to broad categories of mental illness (e.g., other psychiatric disorders, other clients). Mental disorders that were not present in any of the studies were sexual dysfunctions, sleep-wake disorders, gender dysphoria, and elimination disorders.

**Relative stigma, component relations, and other variables.** In contrast to articles on the stigmatization of either mental illness in general or one mental disorder, fewer studies were conducted on stigmatization as it varies with the type of mental illness. In other words, there was a comparatively small amount of research on the relative stigma of mental illness, with just 19.29% of articles exploring this phenomenon. Further, most of the studies on relative stigma did not compare a wide range of mental disorders. Specifically, 55.26% of studies compared two disorders, 23.68% compared three, 18.42% compared four, one article compared six, and one study compared nine. Adding to this, many of the included mental disorders appeared in only a few articles, and a large proportion of the studies included the same two mental disorders, schizophrenia spectrum disorders and depressive disorders. The former was compared to a minimum of one other mental disorder in 65.79% of studies, and the latter was included in 63.16% of articles. Excluding the mental disorders already mentioned at the end of the previous subsection, mental disorders that did not appear in any of the research on relative stigma were somatic symptom disorders, impulse control disorders, and neurocognitive disorders. As

a final observation regarding the literature on relative stigma, many of the quantitative studies did not utilize statistical modelling either at all or in the appropriate manner (i.e., performing an ANOVA without following up with multiple comparisons). In 36.36% of the relevant articles, these statistical limitations were apparent for all mental disorder comparisons, and for 12.12% of studies, this occurred for at least one comparison.

Amongst the literature, research on how the components of stigmatization relate to each other was even more scarce than studies on the relative stigma of mental illness, as only 10.66% of articles examined component relations. A number of these studies investigated the effect of stereotypes on emotions and behaviors, whereas there were fewer articles on the relationship between emotions and behaviors. There were also some studies that explored how different stereotypes relate to each other, and one article examined the link between different emotions. However, not one study investigated how the different components of stigmatization relate to each other as a whole in one framework (e.g., a model with emotions mediating the relationship between stereotypes and behaviors). Despite the little amount of research on relative stigma and component relations, a substantial quantity of articles detailed research on other predictors of stigmatization. The proportion of studies that included these other predictor variables was 61.91%. Throughout and frequently within these articles there was an enormous variety of different predictors. The majority of these variables were individual differences (e.g., ethnicity, marital status, trait authoritarianism), while a small number of studies explored situational and other variables (e.g., the use of labels, target sex, target age; media influence). The four most common individual differences were profession, sex, age, and years of professional experience, and the levels of some predictor variables were not specified (e.g., occupational characteristics, professional function). Additionally, although to a lesser extent, several articles that quantitatively examined these other predictors exhibited the same statistical weaknesses present in the research on relative stigma to some degree. Whether relative stigma, component relations, or other predictor variables were being investigated, there were very few studies that did so while explicitly controlling for other variables (i.e., were not merely controlled for as an incidental part of multiple regression analysis).

**The state of evidence.** Within the corpus of articles, there was certainly a lot of variability with respect to particular data items. This was especially the case for the measures and variables, and even within categories that ought to be homogeneous, this level of variability was still evident. As a result of this, it was extremely difficult to identify any general findings amongst the literature, and practically impossible to summarize findings for all of the included variables. To increase the feasibility of outlining broad findings, an overview was attempted only for the most prevalent research questions. However, even after narrowing down the research questions, the level of variability between articles was still large enough to hinder the identification of general findings. Further, on examining these studies numerous inconsistencies were found between the articles regarding both the presence and direction of effects.

**Bibliometric analysis.** Scopus was unable to access 13 articles, and this meant that 184 or 93.40% of studies were included in the direct citation analysis. Of these articles, VOSviewer identified 57 clusters. 32 of the clusters were individual studies that either did not or were not cited by any other study, and this represented 17.39% of all articles in the direct citation analysis. These articles can be seen in Fig 2 as the individual nodes that form most of the belt of studies surrounding the central structure of articles.

The belt of studies in Fig 2 also contains small clusters of two or more interconnected articles that are isolated from all other studies. Together, these small, isolated clusters account for 8.15% of articles. The remaining 137 or 74.46% of studies are all contained within the central structure articles (see Fig 3 for a more detailed image of the structure). Although this structure is a dense network of interconnected studies, it also includes clusters of articles that are mostly

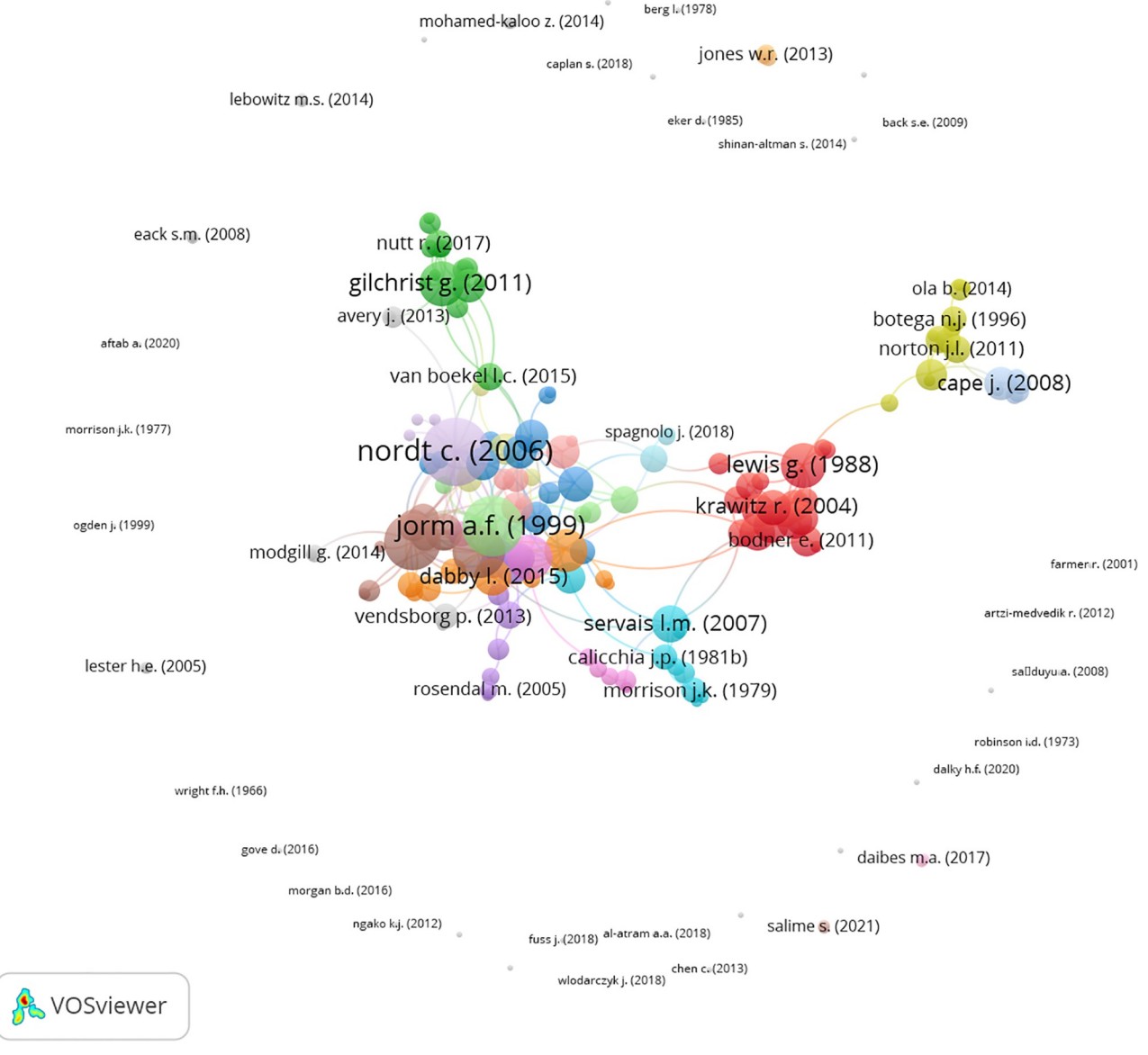

**Fig 2. Global citation network.**

disconnected from the rest of the network. The best examples of this are the green cluster at the top of the network, and the red, yellow, and blue clusters to the right of the network. As depicted in Fig 3, these peripheral clusters are only linked to the remainder of the network indirectly via one or a few other articles. The other studies in the central structure of articles make up the clusters that are closest to each other. These clusters contain the three authors that were cited the most within the whole citation network. Beginning with the most references, Nordt et al. [23] had 27 citations, Jorm et al. [48] received 22 references, and Lauber et al. [49] had 19 citations.

Out of 197 articles, 27 or 13.64% were not found through the scimagojr webpage. For all other studies, the highest and most recent rank in the majority of cases was either Q1 or Q2. In particular, 52.76% of articles were published in Q1 journals, and 34.12% of studies were

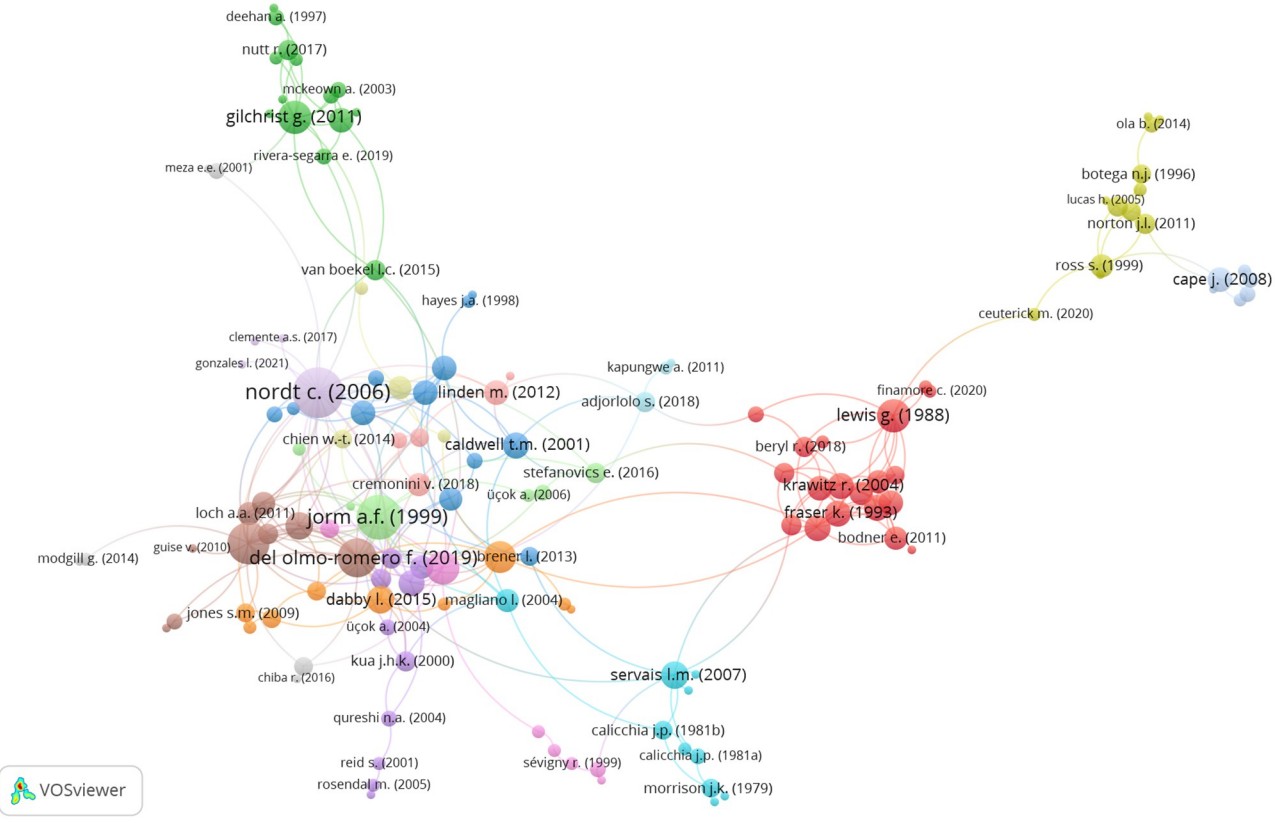

**Fig 3. Central structure of articles.**

published in journals with a rank of Q2. In contrast, the highest and most recent rank for 11.76% of articles was Q3, and for 2.36% of studies, it was Q4.

## Discussion

It was imperative that a scoping review was performed with the aim of outlining the state of available research on the endorsed stigmatization of mental illness by mental health professionals. Most of the objectives for this scoping review were concerned with exploring how studies are being conducted, and whether there are any gaps in the research. Of particular importance was the extent to which studies had been carried out on the relative stigma of mental illness and the components of stigmatization as they relate to each other. As part of a bibliometric analysis, this scoping review also investigated how often articles are connected via a citation, and whether research is being published in well-known journals.

### Summary and interpretation of the results

**Inappropriate samples.** The first major finding of this scoping review was that a substantial proportion of the literature reported results that were based on inappropriate samples. As a result, a subset of the literature was excluded from the subsequent results.

**Gaps in the research.** Another key finding of this scoping review was that there are a number of gaps in the literature that constitute weaknesses in the research on mental health professionals stigmatizing mental illness. Several gaps that were particularly salient concerned

the relative stigma of mental illness and the components of stigmatization. Many studies examined the stigmatization of either mental illness in general or one mental disorder, and in contrast, much less research was conducted on the relative stigma of mental illness. Adding to this, the construct of relative stigma was often not adequately covered. Most of the studies on relative stigma did not compare a variety of mental disorders, and mainly included schizophrenia spectrum disorders and depressive disorders.

When inspecting the dimensions of stigmatization that were investigated, it was found that the emotional and behavioral components of stigmatization were explored in far fewer studies than stereotypes, and this was especially the case for emotions. This lack of research on emotion is striking when considering there is ample evidence to suggest that emotional processes have a stronger relationship with interpersonal contact than stereotypes [50]. This is relevant to mental health practice, as contact with others is an integral part of practice within the mental health field. In addition to little research on emotions and behavior, studies on how the components of stigmatization relate to each other were lacking. Although some studies explored the impact of stereotypes on emotions and behavior, again there was less research investigating the effect of emotions on behavior, and not a single study was conducted on how the components of stigmatization relate to each other as a whole.

Rather than examining the relative stigma of mental illness or component relations, a large proportion of the studies on mental health professionals stigmatizing mental illness were on individual differences (e.g., profession, sex, age). Together with the findings of the previous paragraph, this shows that research in this area has moved little beyond mental illness in general, single mental disorders, stereotypes, and individual differences. This suggests that the literature on mental health professionals stigmatizing mental illness is often not informed by existing theory. However, more importantly, the current state of research on this topic does not supply much guidance on how to reduce this stigmatization and improve the related aspects of mental health practice.

Multiple other gaps were found in the literature on the stigmatization of mental illness by mental health professionals. For example, counselors and occupational therapists participated in only a small proportion of studies, indicating that knowledge in this domain is far from complete for these professional groups. Also, none of the qualitative studies in this area employed theory-building analyses (e.g., grounded theory), and most of the research presented mental illness to participants via labels. The literature would likely benefit from qualitative research that can supply guidance on the variables and hypotheses that warrant examination, and studies that use vignettes or something similar.

**Other limitations.** On top of the limitations noted in the previous subsections, several other limitations were found within the literature on mental health professionals stigmatizing mental illness. These included issues with construct validity, a lack of clarity surrounding aspects of method and results, statistical weaknesses, and inconsistencies with measures and variables. The diversity of measures and variables is more evidence to indicate that established theory is seldom drawn upon and may suggest that variables are frequently selected ad hoc. However, collectively, these limitations likely make it hard to be confident in and comprehend individual articles and the research in general.

**The state of evidence and bibliometric analysis.** Due to the high level of variability and inconsistency of data items, it was deemed mostly infeasible to derive broad findings from the literature on mental health professionals stigmatizing mental illness. As a result, it is unlikely that research on the stigmatization of mental illness by mental health professionals is in a position to direct the reduction of this stigmatization, whether in general or with respect to practice. Additionally, given the nature of studies on mental health professionals stigmatizing

mental illness, it would be difficult if not impossible to conduct either systematic reviews or meta-analyses in this domain.

In prior subsections, it was proposed that variability in the literature and research not moving far beyond mental illness in general, single mental disorders, and individual differences, could be accounted for by studies not drawing upon existing theory. Another possible explanation for this could be that research is disconnected, and the authors of articles on mental health professionals stigmatizing mental illness are not considering other studies in the area. Empirical support for the literature being fragmented in this way was obtained via a direct citation analysis, which showed that close to half of the literature is comprised of individual studies and small systems of research that are isolated to some extent from the rest of the literature. Further, clusters of interconnected articles that were either completely or predominantly disconnected from the remainder of the research were often characterized by a theme (e.g., studies on just borderline personality disorder or personality disorders in general). These results may be due to authors not knowing that research on specific mental disorders or professions sits within a broader literature on mental health professionals stigmatizing mental illness. A third potential explanation for the current state of research could be that studies are not being published in well-known journals. However, it was found that a large proportion of the research was published in Q1 and Q2 journals.

## Recommendations

Based on the findings of this scoping review, recommendations are made regarding prospective research on the endorsed stigmatization of mental illness by mental health professionals.

To reduce stigmatization and improve the relevant facets of mental health practice, a comprehensive understanding of this type of stigmatization needs to be achieved. Namely, studies must focus on the relative stigma of mental illness, multiple dimensions of stigmatization, and how all three components of stigmatization relate to each other in one framework. Moreover, the construct of relative stigma should be sufficiently covered by addressing a range of mental disorders, and research should include all types of mental health professionals.

In order for conclusions to be easily and confidently drawn from this research, studies need to employ procedures that are standard of all good research. However, the current review has shown that this standard of research is often not reached, and studies are not being conducted with a systematic approach. Thus, to have high-quality research the time has come to address studies on the stigmatization of mental illness by mental health professionals programmatically. In particular, many of the limitations within this area of research can likely be overcome with multisite studies.

## Limitations of the current scoping review

While the search string for the current scoping review was far more comprehensive than that of previous literature reviews, a number of studies may have still been overlooked. More specifically, it was impractical to include all classified mental disorders in the search string, and as such articles on mental disorders not in the search string may have been missed in the search. Although, the search string did in fact yield studies on mental disorders that were not part of the search string (e.g., borderline personality disorder, somatic symptom disorder). Another limitation of the current review was that only one reviewer selected the sources of evidence. Consequently, the inclusion and exclusion of articles may not have been entirely consistent with the eligibility criteria. However, an evaluation of inter-reviewer consistency with a second reviewer found a high level of agreement between reviewers, and disagreement led to refinement of the eligibility criteria. A final limitation of the current scoping review pertains to the

direct citation analysis. In this analysis, 13 studies were unable to be included due to these articles not being accessible through Scopus. This means that the citation network produced by the direct citation analysis may have looked different with these studies incorporated, though it is unlikely that the overall findings of this analysis would have changed.

## Conclusion

Mental health professionals, including clinical psychologists, have been found to endorse stigmatizing reactions towards mental illness [5, 8, 22–24]. Despite the importance of research in this domain, the current scoping review found that literature on mental health professionals stigmatizing mental illness is marked by an array of limitations. Notably, multiple gaps were found within the research and the literature has become stagnant. Amongst the gaps in research was a dearth of studies thoroughly exploring the relative stigma of mental illness and how the components of stigmatization relate to each other. As a result of some of these limitations, novel broad findings were unable to be decerned from the literature, and thus research on mental health professionals stigmatizing mental illness is likely not capable of informing the reduction of this stigmatization. It was suggested that several of the limitations in this area may be explained by researchers not drawing on relevant theory and the finding that literature on this topic is partially disconnected. In accordance with the findings of this scoping review, recommendations were proposed for future research on the endorsed stigmatization of mental illness by mental health professionals.

## Supporting information

**S1 Appendix. Table notes and abbreviations.**
(DOCX)

**S2 Appendix. Table A-B.**
(DOCX)

**S3 Appendix. Table C-D.**
(DOCX)

**S4 Appendix. Table E-F.**
(DOCX)

**S5 Appendix. Table G-H.**
(DOCX)

**S6 Appendix. Table I-J.**
(DOCX)

**S7 Appendix. Table K-L.**
(DOCX)

**S8 Appendix. Table M-N.**
(DOCX)

**S9 Appendix. Table O-P.**
(DOCX)

**S10 Appendix. Table Q-R.**
(DOCX)

**S11 Appendix. Table S-T.**
(DOCX)

**S12 Appendix. Table U-V.**
(DOCX)

**S13 Appendix. Table W-Z.**
(DOCX)

**S14 Appendix. Direct citation analysis data.**
(CSV)

**S15 Appendix. Journal type and rank data.**
(XLSX)

**S16 Appendix. PRISMA-ScR fillable checklist.**
(DOCX)

## Author Contributions

**Conceptualization:** Michael Jauch, Stefano Occhipinti, Analise O'Donovan.

**Data curation:** Michael Jauch, Stefano Occhipinti.

**Formal analysis:** Michael Jauch.

**Investigation:** Michael Jauch.

**Methodology:** Michael Jauch, Stefano Occhipinti.

**Project administration:** Michael Jauch, Stefano Occhipinti.

**Resources:** Michael Jauch.

**Software:** Michael Jauch.

**Supervision:** Stefano Occhipinti, Analise O'Donovan.

**Validation:** Michael Jauch, Stefano Occhipinti, Analise O'Donovan.

**Visualization:** Michael Jauch.

**Writing – original draft:** Michael Jauch.

**Writing – review & editing:** Michael Jauch, Stefano Occhipinti, Analise O'Donovan.

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
