## [Decision Letter · Decision Letter 0]

24 Aug 2022

PONE-D-22-02618The Stigmatization of Mental Illness by Mental Health Professionals: Scoping Review and Bibliometric AnalysisPLOS ONE

Dear Dr. Jauch,

Thank you for submitting your manuscript to PLOS ONE. After careful consideration, we feel that it has merit but does not fully meet PLOS ONE’s publication criteria as it currently stands. Therefore, we invite you to submit a revised version of the manuscript that addresses the points raised during the review process.

Please note that we have only been able to secure a single reviewer to assess your manuscript. We are issuing a decision on your manuscript at this point to prevent further delays in the evaluation of your manuscript. Please be aware that the editor who handles your revised manuscript might find it necessary to invite additional reviewers to assess this work once the revised manuscript is submitted. However, we will aim to proceed on the basis of this single review if possible.

We look forward to receiving your revised manuscript.

Kind regards,

Callam Davidson

Editorial Office

PLOS ONE

Journal Requirements:

Additional Editor's comment:

Authors must also state in their “Methods” section whether a protocol exists for their scoping review, and if so, provide a copy of the protocol as supporting information and provide the registry number in the abstract.

Thank you for providing a completed PRISMA checklist. Please update the checklist to use section and paragraph numbers, rather than page numbers. Please add the following statement, or similar, to the Methods: "This study is reported as per the Preferred Reporting Items for Systematic Reviews and Meta-Analyses extension for scoping reviews (PRISMA-ScR) guideline (S16 Appendix)."

Reviewers' comments:

Reviewer's Responses to Questions

**Comments to the Author**

1. Is the manuscript technically sound, and do the data support the conclusions?

Reviewer #1: Yes

2. Has the statistical analysis been performed appropriately and rigorously? 

Reviewer #1: N/A

3. Have the authors made all data underlying the findings in their manuscript fully available?

Reviewer #1: Yes

4. Is the manuscript presented in an intelligible fashion and written in standard English?

Reviewer #1: Yes

5. Review Comments to the Author

Reviewer #1: Thank you very much for giving me the opportunity to review this interesting article.

This is a comprehensive work that deals in depth with the state of research on the stigmatization of health professionals towards people with mental illness.

The manuscript is relevant, it is well structured and the methodology is adequate for the stated objectives. The authors carry out an in-depth analysis and have handled a large amount of information very well.

In fact, I believe that few comments can be made on the content of the manuscript, so my comments are more aimed at clarifying or improving the wording.

- I think the objective section is too long, and much of what is said in it can be said in previous sections. In order to facilitate reading, I recommend that authors in this section limit themselves to describing the objective or objectives as clearly as possible, in a concise and direct manner. To justify the objective, the previous sections must be used.

- Were the searches carried out only in the Title/Abstract fields of the databases, or also in controlled languages (Thesaurus, Mesh...), what do the authors mean when they say that they searched for keywords?

- The process of selecting articles for the review is confusing. It is understood that the search was done in three moments (it is a good idea to update the search if a long time has passed since the first time it was done), but it is difficult to understand, as it is currently written, the total number of records found, the eliminated, etc, since in figure 1 (and in the text) only data from the first phase are provided. In fact, according to the figure, 184 works are included, then it is said that 300, later than 197... In short, I recommend the authors to simplify the explanation given for this process, including the total data of the three phases.

- In the final table of Fig 1 the authors refer to "qualitative synthesis". I have doubts about whether what is being done is a qualitative synthesis. I think it is better to say "articles included in the scoping review"

- On page 23, line 504-507 the authors refer to the variability of the studies and the difficulties in comparing them and obtaining global results. Do the authors believe that this may be related to the breadth of inclusion criteria for their review (multiple professionals, various types of disorders, different types of studies, both qualitative and quantitative)? Perhaps in discussion or in limitations they could include some comment about it.

- The beginning of the discussion (pg 25 lines 546-565) is very long and is repetitive with the introduction. It is true that the introduction should begin with a brief reminder of the objective, but repetition of ideas should be avoided. I recommend making it shorter, remembering the aim and objectives of the study.

- A bibliometric analysis (not scoping review) on stigma in nursing professionals and students (exclusively) has recently been published (https://doi.org/10.3390/ijerph19031839). Although it is not exactly the same objective as this manuscript, I think it could be interesting for the authors to take it into account and perhaps compare results.

6. PLOS authors have the option to publish the peer review history of their article (what does this mean?). If published, this will include your full peer review and any attached files.

Reviewer #1: No

---

## [Author Response · Author response to Decision Letter 0]

6 Sep 2022

Additional requirements

We have ensured that the manuscript meets PLOS One’s style requirements.

Editors’ comments

1. Thank you for providing a completed PRISMA checklist. Please update the checklist to use section and paragraph numbers, rather than page numbers. Please add the following statement, or similar, to the Methods: "This study is reported as per the Preferred Reporting Items for Systematic Reviews and Meta-Analyses extension for scoping reviews (PRISMA-ScR) guideline (S16 Appendix)."

The statement "This study is reported as per the Preferred Reporting Items for Systematic Reviews and Meta-Analyses extension for scoping reviews (PRISMA-ScR) guideline (S16 Appendix)." has been added to the Method section, and the PRISMA checklist has been updated to use section and paragraph numbers (the updated checklist has been attached as an additional file). As PLOS One style requirements do not include section numbers, the section numbers in our PRISMA checklist refer to each of the major manuscript sections in chronological order (i.e., the title page is section 1, the abstract is section 2, the introduction is section 3 etc.) and paragraph numbers refer to each paragraph in chronological order within each section. We apologise for these oversights.

Reviewer’s comments

1. I think the objective section is too long, and much of what is said in it can be said in previous sections. In order to facilitate reading, I recommend that authors in this section limit themselves to describing the objective or objectives as clearly as possible, in a concise and direct manner. To justify the objective, the previous sections must be used.

The Objectives section has been shortened and justifications for the objectives have been moved to the Rationale section.

2. Were the searches carried out only in the Title/Abstract fields of the databases, or also in controlled languages (Thesaurus, Mesh...), what do the authors mean when they say that they searched for keywords?

The searches were only carried out on the title/abstract/keyword fields of the databases. The search was not conducted in controlled languages. The number of results generated from the searches that we carried out on the title/abstract/keyword fields was quite large. Thus, a search for each of the search string terms in controlled languages would have produced an infeasible number of results to screen. 

3. The process of selecting articles for the review is confusing. It is understood that the search was done in three moments (it is a good idea to update the search if a long time has passed since the first time it was done), but it is difficult to understand, as it is currently written, the total number of records found, the eliminated, etc, since in figure 1 (and in the text) only data from the first phase are provided. In fact, according to the figure, 184 works are included, then it is said that 300, later than 197... In short, I recommend the authors to simplify the explanation given for this process, including the total data of the three phases.

We agree with this suggestion and thank the reviewer for this feedback. The figure has been changed to cover all articles included in the scoping review. The figure now includes articles identified through the reference lists of relevant articles from the first database searches and articles identified through the second and third databases searches. Within text, the figure citation was also moved further down to reflect these changes. The distinction between the 300 articles included in the scoping review and the 197 that were isolated in the Results section is discussed in text. We believe that addressing this in text is appropriate.

4. In the final table of Fig 1 the authors refer to "qualitative synthesis". I have doubts about whether what is being done is a qualitative synthesis. I think it is better to say "articles included in the scoping review"

The figure has been changed to reflect this suggestion.

5. On page 23, line 504-507 the authors refer to the variability of the studies and the difficulties in comparing them and obtaining global results. Do the authors believe that this may be related to the breadth of inclusion criteria for their review (multiple professionals, various types of disorders, different types of studies, both qualitative and quantitative)? Perhaps in discussion or in limitations they could include some comment about it.

We agree that it is useful to clarify these distinctions and we have included the following sentence in the Results “Even within categories that ought to be homogeneous this level of variability was still evident”.

6. The beginning of the discussion (pg 25 lines 546-565) is very long and is repetitive with the introduction. It is true that the introduction should begin with a brief reminder of the objective, but repetition of ideas should be avoided. I recommend making it shorter, remembering the aim and objectives of the study.

The beginning of the Discussion has been shortened and now focuses on the aim and objectives of the study.

7. A bibliometric analysis (not scoping review) on stigma in nursing professionals and students (exclusively) has recently been published (https://doi.org/10.3390/ijerph19031839). Although it is not exactly the same objective as this manuscript, I think it could be interesting for the authors to take it into account and perhaps compare results.

We thank the reviewer for pointing out this additional reference. We have included this reference in the Objectives section. Although the results were less directly comparable to the present work, we note the similarity of purpose and increasing use of bibliometric analyses in the field of mental illness stigma.

---

## [Decision Letter · Decision Letter 1]

31 Oct 2022

PONE-D-22-02618R1The Stigmatization of Mental Illness by Mental Health Professionals: Scoping Review and Bibliometric AnalysisPLOS ONE

Dear Dr. Jauch,

Thank you for submitting your manuscript to PLOS ONE. After careful consideration, we feel that it has merit but does not fully meet PLOS ONE’s publication criteria as it currently stands. Therefore, we invite you to submit a revised version of the manuscript that addresses the points raised during the review process.

We look forward to receiving your revised manuscript.

Kind regards,

Juan Diego Ramos-Pichardo, Ph.D.

Guest Editor

PLOS ONE

Reviewers' comments:

Reviewer's Responses to Questions

**Comments to the Author**

1. If the authors have adequately addressed your comments raised in a previous round of review and you feel that this manuscript is now acceptable for publication, you may indicate that here to bypass the “Comments to the Author” section, enter your conflict of interest statement in the “Confidential to Editor” section, and submit your "Accept" recommendation.

Reviewer #2: (No Response)

Reviewer #3: (No Response)

2. Is the manuscript technically sound, and do the data support the conclusions?

Reviewer #2: Partly

Reviewer #3: Partly

3. Has the statistical analysis been performed appropriately and rigorously? 

Reviewer #2: N/A

Reviewer #3: Yes

4. Have the authors made all data underlying the findings in their manuscript fully available?

Reviewer #2: (No Response)

Reviewer #3: Yes

5. Is the manuscript presented in an intelligible fashion and written in standard English?

Reviewer #2: Yes

Reviewer #3: Yes

6. Review Comments to the Author

Reviewer #2: (No Response)

Reviewer #3: Dear authors

It was with great pleasure that I read your manuscript on The Stigmatization of Mental Illness by Mental Health Professionals: Scoping Review and Bibliometric Analysis.

The objective is clear. The introduction and the theoretical rationale justify the relevance of your study. The method is comprehensive regarding scoping review, however the methodological procedures related to bibliometric analysis are very brief. This aspect can be improved, to enrich your manuscript.

Please consult the following references that can help you in writing the method:

Bhandari, A. (2022). Design Thinking: from Bibliometric Analysis to Content Analysis, Current Research Trends, and Future Research Directions. Journal of the Knowledge Economy, 1-56.

Donthu N, Kumar S, Mukherjee D, Pandey N, & Lim W M (2021a). How to conduct a bibliometric analysis: An overview and guidelines. Journal of Business Research, 133(1), 285-296.

Donthu, N., Kumar, S., Pandey, N., & Lim, W.M. (2021b). Research constituents, intellectual structure, and collaboration patterns in Journal of International Marketing: An analytical retrospective. Journal of International Marketing, 29(2), 1-25.

The results of the scoping review and the respective discussion are comprehensive and adequate. The limitations and recommendations are congruent with the results.

Best Regards

7. PLOS authors have the option to publish the peer review history of their article (what does this mean?). If published, this will include your full peer review and any attached files.

Reviewer #2: No

Reviewer #3: **Yes: **Luís Sousa

---

## [Author Response · Author response to Decision Letter 1]

14 Nov 2022

November 15, 2022

Subject: Submission Revised Manuscript ID PONE-D-22-02618

Dear Editor of PLOS One

Please find attached our revised manuscript for submission. We are pleased that our previous revisions have addressed most of the reviewers’ concerns. We list in point form below our response to the remaining issues identified.

Reviewer 2

1. The objectives section is still too long. It is not necessary to provide so much information as it is already provided in the previous sections.

The Objectives section has been further shortened to exclude unnecessary information.

2. In the section Populations, time periods, and the extent of research results a lot of information previously given is repeated. Lines 558 to 570 of the section summary and interpretation of the reults are repeated in the section Populations, time periods, and the extent of research. You have already provided the data in the previous section in an exhaustive way. I believe that indicating only the general results simplifies the reading.

Most of this section has been removed. This section now only includes sentences about inappropriate samples and about excluding a proportion of the literature. Consequently, the subheading for this section has been changed to Inappropriate Samples.

3. The bibliometric analysis. Some of the results are repeated in the discussion without justification. Direct citation analysis is used to determine the impact of particular articles or authors. Therefore, I think that the work would be improved if you would describe in more detail the results obtained: how many clusters you have obtained, the main authors cited, indicating the number of citations and the clusters that are closest to each other. In this way, in the discussion you can try to find patterns in the use of scientific information by groups of authors or if collaboration is scarce on certain topics. In relation to the above, the paragraph on lines 536 -538 "clusters were often characterised by a theme. For instance, the aforementioned red cluster contained studies that were all about either borderline personality disorder or personality disorders in general" is not a direct result. You should move it to the discussion and justify with the authors' citations why you come to that conclusion or why you think the authors investigate similar themes.

The results of the direct citation analysis have been described in more detail as suggested by the reviewer (i.e., how many clusters you have obtained, the main authors cited, indicating the number of citations, and the clusters that are closest to each other). Interpretations about the themes of different clusters have been removed from the Results. Given these interpretations were already in the Discussion, they have remained in this section. As well, the points in this paragraph have been reordered for clarity.

4. The discussion section in general is too long and often repeats information without providing interpretation or comparison with other reviews. It would improve the readability if it were shorter.

In general, the Discussion section has been shortened to exclude unnecessary repetition of the results.

Reviewer 3

1. The methodological procedures related to bibliometric analysis are very brief. This aspect can be improved, to enrich your manuscript. Please consult the following references that can help you in writing the method:

Bhandari, A. (2022). Design Thinking: from Bibliometric Analysis to Content Analysis, Current Research Trends, and Future Research Directions. Journal of the Knowledge Economy, 1-56.

Donthu N, Kumar S, Mukherjee D, Pandey N, & Lim W M (2021a). How to conduct a bibliometric analysis: An overview and guidelines. Journal of Business Research, 133(1), 285-296.

Donthu, N., Kumar, S., Pandey, N., & Lim, W.M. (2021b). Research constituents, intellectual structure, and collaboration patterns in Journal of International Marketing: An analytical retrospective. Journal of International Marketing, 29(2), 1-25.

We thank the reviewer for these suggestions, and we have consulted these sources. We see that our analysis is largely consistent with the suggested approach. However, we have brought together information about the analyses into the section of the Method about our bibliometric analysis. This, in conjunction with extra material suggested by other reviewers, makes clearer the steps in the analysis for the reader. 

Once again, we would like to thank the editor and reviewers for the opportunity to revise this paper and for considering it for publication. We look forward to hearing of the outcome.

Kind Regards,

The Authors

---

## [Decision Letter · Decision Letter 2]

20 Dec 2022

PONE-D-22-02618R2The Stigmatization of Mental Illness by Mental Health Professionals: Scoping Review and Bibliometric AnalysisPLOS ONE

Dear Dr. Jauch,

Thank you for submitting your manuscript to PLOS ONE. After careful consideration, we feel that it has merit but does not fully meet PLOS ONE’s publication criteria as it currently stands. Therefore, we invite you to submit a revised version of the manuscript that addresses the points raised during the review process.

We look forward to receiving your revised manuscript.

Kind regards,

Juan Diego Ramos-Pichardo, Ph.D.

Guest Editor

PLOS ONE

Journal Requirements:

Additional Editor Comments:

Thank you very much for the effort you put into responding to all reviewer comments.

The manuscript has improved considerably, however there are still some suggestions that must be addressed before acceptance.

Please, I beg you to take into account the comments of the reviewers and to resubmit a new version of the manuscript.

Best regards

Reviewers' comments:

Reviewer's Responses to Questions

**Comments to the Author**

1. If the authors have adequately addressed your comments raised in a previous round of review and you feel that this manuscript is now acceptable for publication, you may indicate that here to bypass the “Comments to the Author” section, enter your conflict of interest statement in the “Confidential to Editor” section, and submit your "Accept" recommendation.

Reviewer #2: (No Response)

Reviewer #3: All comments have been addressed

2. Is the manuscript technically sound, and do the data support the conclusions?

Reviewer #2: Yes

Reviewer #3: Yes

3. Has the statistical analysis been performed appropriately and rigorously? 

Reviewer #2: N/A

Reviewer #3: N/A

4. Have the authors made all data underlying the findings in their manuscript fully available?

Reviewer #2: Yes

Reviewer #3: Yes

5. Is the manuscript presented in an intelligible fashion and written in standard English?

Reviewer #2: Yes

Reviewer #3: Yes

6. Review Comments to the Author

Reviewer #2: Congratulations to the authors. The work is much better structured, which makes it easier to read. The bibliometric analysis has improved qualitatively, providing results that can be analysed and conclusions drawn.

I have a few suggestions.

In the objectives section, line 154 and 155, it says: "The aim of this review is to examine the state of all available research on the stigmatisation of mental illness by mental health professionals".

I would change the word "all avaiable" for another expression that is not so absolute.

In the results you indicate in lines 522 to 525:

Furthermore, in examining these studies, numerous inconsistencies were found between the articles with regard to both the presence and direction of effects. In fact, the only overall finding that was clearly discernible was that mental health professionals endorse both positive and stigmatising reactions to mental illness.

However, the conclusions seem to indicate that the results indicate that professionals have stigmatising reactions to mental illness. Lines 774 and 775

Mental health professionals, including clinical psychologists, were found to have

stigmatising reactions to mental illness (5, 21-24).

If the main result is that it is not possible to have a clear conclusion whether professionals have positive or negative reactions it should be reflected in the findings.

Reviewer #3: Dear Authors

The manuscript became clearer after revision.

The manuscript "The Stigmatization of Mental Illness by Mental Health Professionals: Scoping Review and Bibliometric Analysis" presents the clear objective, method and results as well as the discussion are adequately done.

Continuation of a good work

Best Regards

7. PLOS authors have the option to publish the peer review history of their article (what does this mean?). If published, this will include your full peer review and any attached files.

Reviewer #2: No

Reviewer #3: **Yes: **Luís Sousa

---

## [Author Response · Author response to Decision Letter 2]

21 Dec 2022

December 22, 2022

Subject: Submission Revised Manuscript ID PONE-D-22-02618

Dear Editor of PLOS One

Please find attached our revised manuscript for submission. We are pleased that our previous revisions have addressed most of the reviewers’ concerns. We list in point form below our response to the remaining issues identified.

Reviewer 3

1. In the objectives section, line 154 and 155, it says: "The aim of this review is to examine the state of all available research on the stigmatisation of mental illness by mental health professionals". I would change the word "all avaiable" for another expression that is not so absolute.

We thank the reviewer for this feedback. We have removed the word “all” from the phrase “all available” in both the Objectives section and Discussion section. The aim now reads “to examine the state of available research on the endorsed stigmatization of mental illness by mental health professionals”.

2. In the results you indicate in lines 522 to 525: Furthermore, in examining these studies, numerous inconsistencies were found between the articles with regard to both the presence and direction of effects. In fact, the only overall finding that was clearly discernible was that mental health professionals endorse both positive and stigmatising reactions to mental illness. However, the conclusions seem to indicate that the results indicate that professionals have stigmatising reactions to mental illness. Lines 774 and 775 Mental health professionals, including clinical psychologists, were found to have stigmatising reactions to mental illness (5, 21-24). If the main result is that it is not possible to have a clear conclusion whether professionals have positive or negative reactions it should be reflected in the findings.

We understand the reviewer’s confusion with the sentence “In fact, the only overall finding that was clearly discernible was that mental health professionals endorse both positive and stigmatising reactions to mental illness”. Rather than suggesting there was no effect of stigmatization, we intended to illustrate the ambivalence of some of these findings. However, as this was not a key finding of the present scoping review, and other authors have already noted similar results (i.e., mental health professionals expressing ambivalence towards people with mental illness by endorsing both positive and stigmatising reactions), we have removed this sentence. We feel that this removes a potential source of contradiction for the reader.

Once again, we would like to thank the editor and reviewers for the opportunity to revise this paper and for considering it for publication. We look forward to hearing of the outcome.

Kind Regards,

The Authors

---

## [Decision Letter · Decision Letter 3]

8 Jan 2023

The Stigmatization of Mental Illness by Mental Health Professionals: Scoping Review and Bibliometric Analysis

PONE-D-22-02618R3

Dear Dr. Jauch,

We’re pleased to inform you that your manuscript has been judged scientifically suitable for publication and will be formally accepted for publication once it meets all outstanding technical requirements.

Kind regards,

Juan Diego Ramos-Pichardo, Ph.D.

Guest Editor

PLOS ONE

Additional Editor Comments (optional):

Reviewers' comments:

Reviewer's Responses to Questions

**Comments to the Author**

1. If the authors have adequately addressed your comments raised in a previous round of review and you feel that this manuscript is now acceptable for publication, you may indicate that here to bypass the “Comments to the Author” section, enter your conflict of interest statement in the “Confidential to Editor” section, and submit your "Accept" recommendation.

Reviewer #2: (No Response)

2. Is the manuscript technically sound, and do the data support the conclusions?

Reviewer #2: Yes

3. Has the statistical analysis been performed appropriately and rigorously? 

Reviewer #2: N/A

4. Have the authors made all data underlying the findings in their manuscript fully available?

Reviewer #2: Yes

5. Is the manuscript presented in an intelligible fashion and written in standard English?

Reviewer #2: Yes

6. Review Comments to the Author

Reviewer #2: Congratulations to the authors for their work. I believe that after these latest corrections the results and conclusions show more coherence. For my part, the article is of sufficient quality to be published.

7. PLOS authors have the option to publish the peer review history of their article (what does this mean?). If published, this will include your full peer review and any attached files.

Reviewer #2: No

---

## [Editor Report · Acceptance letter]

12 Jan 2023

PONE-D-22-02618R3 

The stigmatization of mental illness by mental health professionals: Scoping review and bibliometric analysis 

Dear Dr. Jauch:

I'm pleased to inform you that your manuscript has been deemed suitable for publication in PLOS ONE. Congratulations! Your manuscript is now with our production department. 

Kind regards, 

on behalf of

Dr. Juan Diego Ramos-Pichardo 

Guest Editor

PLOS ONE